# Interpretable Structural Evaluation of Metal-Oxide Nanostructures in Scanning Transmission Electron Microscopy (STEM) Images via Persistent Homology

**DOI:** 10.3390/nano14171413

**Published:** 2024-08-29

**Authors:** Ryuto Eguchi, Yu Wen, Hideki Abe, Ayako Hashimoto

**Affiliations:** 1National Institute for Materials Science, 1-2-1 Sengen, Tsukuba 305-0047, Ibaraki, Japan; yu.wen.lw@alumni.tsukuba.ac.jp (Y.W.); abe.hideki@nims.go.jp (H.A.); 2Graduate School of Pure and Applied Sciences, University of Tsukuba, 1-2-1 Sengen, Tsukuba 305-0047, Ibaraki, Japan; 3Graduate School of Science and Engineering, Saitama University, Shimo-Okubo 255, Saitama 338-8570, Japan

**Keywords:** persistent homology, metal-oxide nanostructures, scanning transmission electron microscopy (STEM)

## Abstract

Persistent homology is a powerful tool for quantifying various structures, but it is equally crucial to maintain its interpretability. In this study, we extracted interpretable geometric features from the persistent diagrams (PDs) of scanning transmission electron microscopy (STEM) images of self-assembled Pt-CeO_2_ nanostructures synthesized under different annealing conditions. We focused on PD quadrants and extracted five interpretable features from the zeroth and first PDs of nanostructures ranging from maze-like to striped patterns. A combination of hierarchical clustering and inverse analysis of PDs reconstructed by principal component analysis through vectorization of the PDs highlighted the importance of the number of arc-like structures of the CeO_2_ phase in the first PDs, particularly those that were smaller than a characteristic size. This descriptor enabled us to quantify the degree of disorder, namely the density of bends, in nanostructures formed under different conditions. By using this descriptor along with the width of the CeO_2_ phase, we classified 12 Pt-CeO_2_ nanostructures in an interpretable way.

## 1. Introduction

Metal–oxide nanocomposites with strong interfacial interactions have been studied extensively because their excellent chemical and physical properties make them attractive as catalysts and electrode materials [1,2]. As the structure of metal–oxide nanocomposites affects their performance, investigating the structure–property relationship of nanocomposites can provide information to improve materials design. Traditionally, geometric factors such as size [3,4,5,6], density [7,8], area, and shape are useful for describing the structure of materials quantitatively [9,10]. However, because most nanostructures do not have a regular structure, discussions on their relationships to physical properties based on such geometric factors are limited. Therefore, more suitable geometrical features are required for property predictions.

Topological data analysis (TDA) [11,12] is one of the powerful tools that have the possibility to provide alternative structural features in materials science. With its versatility and time efficiency, TDA has been used to characterize the structure of organic species, proteins [13], inorganic materials, and amorphous materials [14]. Unlike the traditional geometric measurements of the dimensions, TDA can quantify the multi-dimensional and multi-scale topological intrinsic features. The features based on the homology capture the data shape as N-dimensional holes, including connected components as zero-dimensional (0D) holes and loops or tunnels as one-dimensional (1D) holes. The number of N-dimensional holes, called the Betti number (*β*_N_), is known for topological invariants. Using this basic descriptor, it becomes easy to tell the difference between two complicated structures and deduce the quantitative relationship between structure and function. In our previous work [15,16], we applied *β*_0_ and *β*_1_ to scanning transmission electron microscopy (STEM) and STEM tomography for analysis of both the two-dimensional (2D) and three-dimensional (3D) structures of various self-assembled Pt-CeO_2_ nanocomposites that were synthesized under different annealing conditions. The quantitative relationship between the CeO_2_ phase connectivity and oxygen-ion conductivity was successfully established by using *β*_0_ of CeO_2_ phases as a structural descriptor.

Persistent homology (PH) includes specific and local geometric features besides *β*_N_ with rough and global approximations. The main concept of PH lies in tracking the scale required for the appearance and disappearance of N-dimensional holes through continuous deformation. Each homological feature that appears at birth time *b* and disappears at death time *d* is recorded. The birth and death positions of a structure can be identified by inverse analysis. Therefore, PH is a competent approach for structure description and has been applied to many fields, such as amorphous structures [17,18], magnetic domain structures [19,20], and spin systems [21]. In recent years, studies have also predicted material properties using PH-based machine learning. Minamitani and co-workers predicted the thermal conductivity of amorphous carbon [22,23]. Sato et al. [24] evaluated the activation energy of ionic conductivity in silver iodide (AgI) by investigating its atomic configurations constructed with molecular dynamics simulations. Uesugi et al. [25] created the ridge regression model that predicted the pre-exponential factor and activation energy of the Pt-CeO_2_ nanocomposites. Thus, PH successfully generated quantitative geometric features and thereby revealed new relationships to physical properties. Then, the physical meaning of the features should be understood in order to examine the validity of the models for property predictions. However, useful structural descriptors from PH-based machine learning are not always interpretable. Furthermore, when analyzing experimental data for material design, machine learning may be hardly able to be performed or hardly produce expected results due to the limited number of data. The interpretability of structural descriptors would promote revealing the relationships between synthesis conditions, structures, and functions even from the limited data.

In this study, we aimed to extract interpretable structural descriptors to facilitate direct structural classification for the Pt-CeO_2_ nanocomposites from the STEM images by using PH. We could extract the averaged structural features from the PDs of the STEM images. Further step-by-step analysis found new important interpretable features with the characteristic size information. Finally, the two effective descriptors were determined by a random forest classification, which quantified the scale and the degree of disorder, namely the density of bends. Their scatter plots provided simple and interpretable classification. This approach would be able to contribute to identifying the more important relationship between structure and transport properties, such as ionic conductivity.

## 2. Materials and Methods

Preparation of Pt-CeO_2_ nanocomposites, acquisition of STEM images, and pre-processing of STEM images were conducted according to our previous reports [15,16]. Twelve phase-separated Pt-CeO_2_ nanocomposites were synthesized by annealing Pt_5_Ce alloy at different temperatures (500, 600, and 700 °C) and syngas ratios (CO:O_2_ = 0:1, 1:1, 2:1, and 3:1).

The nanostructures of the Pt-CeO_2_ nanocomposites were characterized using high-angle annular dark-field STEM (HAADF-STEM; JEM-2100F, JEOL, Tokyo, Japan). Seven images of each nanocomposite were selected and trimmed to the same size and scale (1024 × 1024 pixels, 406 × 406 nm). The Pt and CeO_2_ phases were identified by their bright and dark contrast, respectively. The STEM images were processed by background removal and binarization (Appendix A).

Figure 1 shows a conceptual diagram for the extraction of interpretable structural features from STEM images by using PH and the classification of nanostructures by using the selected effective features. We extracted the averaged structural features from the PDs of the STEM images, while further step-by-step analysis by vectorization of all the PDs was conducted to add alternative important features with the characteristic size information. As a final target, the effective descriptors were determined by a random forest classification, which was used for the interpretable classification.

To obtain PDs, first, each pixel value in the binarized STEM images was assigned by a signed Euclidean distance (SED) from the boundaries between the black (CeO_2_) and white (Pt) phases. Each pixel was given a sign: negative for the black domains and positive for the white ones (Appendix A). Here, we focused on the homology of the CeO_2_ phase because of its correlation with oxygen ionic conductivity as well as the previous work [15,16]. Second, a filtration was processed to calculate *b* and *d*, where the threshold for binarization increased, persisting from the minimum to the maximum value. When N-dimensional holes appeared and disappeared during the filtration, the SED thresholds were defined as *b* and *d*, respectively. Then, 2D histograms of birth and death pairs (*b*-*d* pairs) were obtained as PDs. The difference between birth and death time (*d* − *b*) is defined as lifetime. We calculated the zeroth PD for 0D holes and the first PD for 1D holes.

Next, we conducted hierarchical clustering and PCA for each vectorized PD to classify the different Pt-CeO_2_ nanostructures and identify important features. The persistence image (ρ) was used for vectorization of PDs [26]. Because a PD consists of a set of *b*-*d* pairs ${\{(b}_{k},d_{k}$): k=1,2,…, l}, where $b_{k}\;\left(d_{k}\right)$ is from the k-th hole and l is the total number of holes, ρ is defined as
(1)$$\rho \left(x,\;y\right)=\sum_{k=1}^{l}w\left(b_{k},d_{k}\right)\exp\left(-\frac{{\left(b_{k}-x\right)}^{2}+{\left(d_{k}-y\right)}^{2}}{2\sigma^{2}}\right),$$
(2)$$w\left(b_{k},d_{k}\right)=arctan\left(C{\left(d_{k}-b_{k}\right)}^{p}\right)$$

Here, the variables C, p, and σ were set as 2.0, 0.5, and 1.0, respectively. We discretized the region [−12, 6] × [−12, 6] in the zeroth PD and [−6, 12] × [−6, 12] in the first PD into 90 × 90 grids and then evaluated ρ (x, y) of the respective grids. Vectorized PDs were calculated by organizing each ρ (x, y) in a prefixed order.

Hierarchical clustering was performed using the cosine similarity and dissimilarity of vectorized zeroth PDs to evaluate the closeness of each PD [27]. We calculated the averaged vectors from the seven vectorized PDs with the same synthesis conditions, resulting in 12 vectors for each set of conditions. Subsequently, cosine dissimilarities for all the combinations of 12 vectors were calculated and used as the matrix distance. A weighted method was used for calculating the distance between the clusters as the averaged cosine dissimilarities for all possible pairs [28]. PCA decreases the dimensionality of datasets and maximizes their variance, thereby allowing us to extract critical information and emphasize the differences between input vectors [29]. A set of 84 vectors (seven images × 12 sets of conditions) was processed for each of the zeroth and first PDs. Furthermore, the PDs were reconstructed with the first principal component (PC1) obtained through PCA by projecting PC1 onto the grid used in the PD vectorization. The procedures for PD calculation, visualization, vectorization, and reconstruction by PCA were conducted using the data analysis software ‘HomCloud’ [30,31].

Finally, the random forest method was used to quantify the importance of PH features in classification tasks [32]. The image sets were augmented to encompass a thirty-sixfold increase in the total image number by segmenting the images into 3 × 3 pieces and rotating these pieces every 90°. We extracted the five interpretable features from the PD quadrants of all augmented images and adapted them as explanatory variables for the classification of four types of sample sets, which were chosen from the result of hierarchical clustering. Thus, we evaluated the importance of the extracted features based on their contribution to the decrease in a Gini impurity, that is, a decrease in the number of misclassifications. The number of trees was set to 100 and the maximum depth of trees was set to five to prevent the overfitting.

## 3. Results and Discussion

### 3.1. Extraction of Interpretable Features from Persistent Diagram Quadrants

Figure 2a–c show binarized HAADF-STEM images of the structural transformation of Pt-CeO_2_ nanostructures with rising formation temperature from 500 to 700 °C. The structure changed from maze-like to striped as temperature increased. Figure 2d–f show the zeroth and Figure 2g–i show the first PDs derived from the CeO_2_ phase in the corresponding nanostructures in Figure 2a–c, respectively. The distribution of individual *b*–*d* pairs changes in the zeroth and first PDs, reflecting the structural changes; however, the quadrants in which the *b*-*d* pairs are located remain the same in both PDs. Therefore, we first focused on the PD quadrants, namely, the signs of *b* and *d* which allowed us to extract five interpretable features. In the zeroth PDs (Figure 2d–f), because new domains can appear only through shrinkage, *b* is inevitably negative and captures the scale required for the appearance of a domain during shrinkage. Hence, for striped domains, the absolute value of *b* almost corresponds to half of the stripe width (Appendix A). In contrast, *d* exhibits both negative and positive signs, which describe the splitting of the connected domains through shrinkage and the merging of isolated domains through expansion, respectively. In the region with positive *d*, the number of *b*-*d* pairs represents the number of isolated domains minus one, which additionally corresponds to the frequency of merging (Appendix A). Thus, we could derive the trends in the width and number of CeO_2_ domains from the positive *d* region. The absolute *b* values, which were estimated by Gaussian fitting as shown in Figure 3a–c, increased with the annealing temperature, directly implying the average width of the CeO_2_ phase increased with temperature. In Figure 3d, the number of the CeO_2_ domains decreased with rising fabrication temperature. These results from the PD analysis coincide with the visual analysis of the STEM images (Figure 2a–c). Additionally, the total lengths of stripes from all domains in one image could be estimated by counting the number of *b*-*d* points with short lifetimes (Appendix A). In the first PDs (Figure 2g–i), because ring-shaped CeO_2_ domains can disappear only through expansion, *d* was exclusively positive and captured the scale required for the merge of ring-shaped domains, which, therefore, could be regarded as the diameter of ring-shaped structures (Appendix A). In the first PDs, *b* was either negative or positive, which described the splitting of the rings through shrinkage and the evolution from arc-like structures to rings through expansion, respectively. Thus, the number of *b*-*d* pairs in the respective regions represents those of rings and arc-like structures (Appendix A).

Table 1 summarizes the relationships between the five interpretable features and PD quadrants. The average and error values for three Pt-CeO_2_ nanostructures fabricated at different temperatures calculated from seven images of structures formed under the same synthesis conditions are also listed in Table 1. The interpretable features of the Pt phase were obtained from the PDs after reversing black and white regions in the binarized images. For width measurements, only *b*-*d* pairs near *b* = *d* were used because they mainly originated from the striped structure rather than particle- and dumbbell-like structures (Appendix A). The total length of the CeO_2_ and Pt stripes from all domains decreased as the synthesis temperature increased. This trend agrees with the actual phase change from maze-like to striped structure observed in the STEM images. The number of arcs in both phases decreased with rising synthesis temperature, quantitatively indicating the decrease of curved structure in the samples. The number of CeO_2_ domains decreased, whereas that of Pt domains remained almost the same as the synthesis temperature increased. Conversely, the number of CeO_2_ rings remained almost the same, whereas the number of Pt rings decreased as the fabrication temperature rose. These findings indicate that Pt inhibits the diffusion of Ce during the structural formation process [33], resulting in the development of many isolated CeO_2_ domains at low fabrication temperatures. At this point, the inhibitor Pt surrounds CeO_2_, thus forming many Pt rings. Conversely, at high synthesis temperatures, where the diffusion rate of Ce is high, it becomes less inhibited by Pt than is the case at lower temperatures, allowing isolated Ce domains to connect and decreasing the number of domains. Under these conditions, Pt no longer easily surrounds CeO_2_, resulting in a decrease in the number of Pt rings.

### 3.2. Visualization of Interpretable Features by Inverse Analysis of Reconstructed Persistent Diagrams Based on Principal Component Analysis

Although five geometric features were extracted by interpreting the PD quadrants, they were averaged or summed values that did not keep the information of each PD. Therefore, all 84 PDs of the Pt-CeO_2_ nanostructures were examined by vectorization to deepen our knowledge of their features. We visualized the interpretable features by a combination of hierarchical clustering and inverse analysis of reconstructed PDs based on PCA. First, all the mean cosine similarities were calculated from the vectorized PDs for hierarchical clustering. The mean cosine similarities exceeded 0.99 within the same set of synthesis conditions, and their maximum deviation was 0.007148 (Appendix A). The high similarities and small deviation implied that the Pt-CeO_2_ nanostructures formed under the same conditions were almost identical. Figure 4a shows the hierarchical clustering with cosine dissimilarities for 12 different nanostructures. At the first branch point, all the nanostructures are divided into two clusters: one cluster (G1–G2–G3) for structures formed using CO:O_2_ gas ratios of 0:1 and 1:1 and the other (G4–G5–G6) for structures formed using CO:O_2_ gas ratios of 2:1 and 3:1, except for the sample fabricated at 500 °C with a CO:O_2_ gas ratio of 2:1. The binarized STEM images in Figure 4b clearly show a structural difference between the two clusters in the stripe width. While the average annealing temperatures for the two groups were calculated to be 586 and 620 °C, the average CO:O_2_ gas ratios of the two groups were 0.714:1 and 2.60:1. The larger relative difference between the gas ratios of the two groups than between temperatures could indicate that gas ratio has a stronger effect on the structure of the materials than temperature. At the second and third branch points, the two clusters were further divided into two at each point. This clustering might be based on temperature differences rather than the gas ratios. However, specific geometric differences are challenging to discern directly from STEM images. After the fifth branch point, the cosine dissimilarities fell within the range of the maximum deviation, i.e., the threshold for clustering, as indicated by the dashed line. Thus, all 12 nanostructures could be clustered into six groups (G1–G6).

Second, an inverse analysis of the PDs reconstructed from PCA was conducted to visualize the details of the structural differences between the groups. In the PCA of the original PDs, the contribution ratios of principal component 1 (PC1) and principal component 2 (PC2) were 0.862 and 0.073 for the zeroth PDs, respectively, and 0.864 and 0.105 for the first PDs, respectively. The large PC1 contribution ratio indicates that PC1 was sufficient to represent the nano-structural differences between the groups. Therefore, we focused on only PC1 and obtained the zeroth and first reconstructed PDs, which are shown in Figure 5a,c, respectively. The *b*-*d* pairs in the red and blue regions contribute positively and negatively to PC1, respectively. Figure 5b,d plot the PC1 values from all 84 vectorized PDs categorized by the six groups G1–G6 through hierarchical clustering (Figure 4) to show how PC1 contributes to the clustering. The PC1 values for both the zeroth and first reconstructed PDs have obvious dependence on each group, which further confirms that PC1 can effectively describe the nanostructures. All 84 PDs of the Pt-CeO_2_ nanostructures were examined by vectorization and we were able to visualize interpretable features by a combination of hierarchical clustering and inverse analysis of reconstructed PDs based on PCA.

Then, the geometric feature related to PC1 was revealed through inverse analysis of the reconstructed PDs, which enabled us to identify the corresponding N-dimensional holes to the red (positive) and blue (negative) regions in the reconstructed PDs (Figure 5a,c). Here, to maintain simplicity and interpretability, we set a characteristic scale of 4 nm to define specific blue and non-blue regions by the characteristic scale of *b* in the zeroth reconstructed PDs and *d* in the first ones because the blue and non-blue regions can be separated by this characteristics value (Appendix A). Furthermore, we confined *b*-*d* pairs with −4 nm ≤
*b*, *d*
< 0 in the zeroth PDs for the connected domains and those with 0 <
*b*, *d*
< 4 nm in the first PDs for arc-like structures, which correspond to the blue (negative) regions. The inset images in Figure 5b,d show the inverse analysis results obtained from the zeroth and first reconstructed PDs for the nanostructure synthesized at 700 °C with a 1:1 syngas ratio and at 700 °C with a 2:1 syngas ratio, which had a large difference in PC1 values. Their blue and red phases correspond to the blue and non-blue regions in the reconstructed PDs, respectively. We can clearly observe that the area fraction of the blue phase decreases as PC1 increases for both PDs. This implies that the difference in hierarchical clustering between the six groups could mainly be explained by the number of narrower domains and smaller arcs rather than the characteristic size. Another important finding here is that the number of small arcs can represent the degree of disorder because it reflects the density of local bends in the nanostructures. The number of domains was interpreted as the connectivity in a previous study [15]. Even though the stripes are maintained, they often bend to form maze-like structures. That is, the number of small arcs can act as an effective feature to describe the nanostructure rather than the number of domains. The visualization of the extracted features by a combination of hierarchical clustering and inverse analysis revealed that considering even the characteristic size of the features made the meaning of the features clearer and improved their interpretability.

### 3.3. Determination and Use of Effective Interpretable Descriptors

A random forest classification was performed to evaluate the five features obtained and determine the most effective descriptors for structural classification as the final objective. Here, from the feature visualization, the number of arcs was replaced by that of small arcs with *d*
< 4 nm. To improve the reliability for evaluation of the feature importance, we made four sample sets consisting of the three nanostructures confirmed to be classifiable by hierarchical clustering (Appendix A). In all the sample sets, the number of small arcs and the width of the CeO_2_ phase were the most and second-most important features, respectively, among the five features used for the random forest classification (Appendix A). Interestingly, the number of small arcs was the most important feature in sample set 1, which consisted of nanostructures fabricated at different temperatures with the same gas ratio (Figure 2). This result is consistent with that of inverse analysis using the PC1 contribution (Figure 5). It was confirmed that binarization procedures with different parameters had little effect on the determination of effective descriptors through random forest classification (Appendix A).

Figure 6 displays a scatter plot of the number of small arcs with the characteristic size and width of the CeO_2_ phases identified as the effective descriptors for 12 Pt-CeO_2_ nanostructures. The scatter plots clearly classify the various nanostructures synthesized under different conditions. The width (vertical axis) behaves like a scale factor, and the number of small arcs (horizontal axis) behaves as a descriptor of disorder. For comparison, the well-classified nanostructures synthesized at 500, 600, and 700 °C with a gas ratio of 2:1 are shown in blue, green, and red, respectively. As observed in the corresponding binarized STEM images (inset), while the nanostructures at the top left (red) are striped, those at the bottom right (blue) are maze-like, with more small arcs present in the samples formed at lower temperatures, as mentioned in Section 3.1. Notably, PCA could also classify the 12 nanostructures but provided little direct interpretability (Appendix A).

In addition, these two simple and interpretable descriptors could be utilized for material design to control the ionic conductivity of Pt/CeO_2_. The lower activation energy should increase ionic conductivity. In our previous measurements, the nanostructures synthesized at 500, 600, and 700 °C with a gas ratio of 2:1 had activation energies of 0.74 eV, 1.2 eV, and 1.24 eV, respectively [16]. Comparing them with the results in Figure 6, the smaller the width and the more the number of small arcs, the lower the activation energy. This trend could indicate that increasing the interfacial area between Pt and CeO_2_ phases with the smaller width and the number of small arcs led to an increase in ionic conductivity. This assumption agrees with the fact that oxygen ions diffuse rapidly at the interface between metal and oxygen [34]. Thus, we could not only classify the nanocomposites but also directly visualize the structural differences, contributing to establishing the guidelines for materials design by the two interpretable descriptors.

## 4. Conclusions

We quantitatively evaluated Pt-CeO_2_ nanostructures ranging from maze-like to striped structures by using the PDs of their STEM images. We aimed to effectively extract interpretable features to enable structural classification. Analysis of the PD quadrants provided five features: average width and total length of the striped CeO_2_ phases, the number of CeO_2_ phases from the zeroth PDs, and the numbers of ring- and arc-like structures from first PDs. The combination of hierarchical clustering and inverse analysis of each vectorized PD revealed that the addition of the characteristic size to the features effectively improved the interpretability; that is, the number of small arc-like structures represented the density of bends in the nanostructures. Finally, the two most important interpretable descriptors among the five features were extracted by a random forest classification: the number of small arcs and the width of the CeO_2_ phase. A simple 2D scatter plot based on these two descriptors could effectively classify the nanocomposites and visualize their structure differences via the two metrics of degree of disorder and scale factor. Thus, PH facilitates the assessment of the structural differences that arise from variations in synthesis conditions/methods. Furthermore, the extraction of the interpretable descriptors in this way was effective and convenient when quantitatively evaluating the relationship between quantitative properties such as ionic conductivity. Using this approach, it is possible to predict properties from structural descriptors.

## Figures and Tables

**Figure 1 nanomaterials-14-01413-f001:**
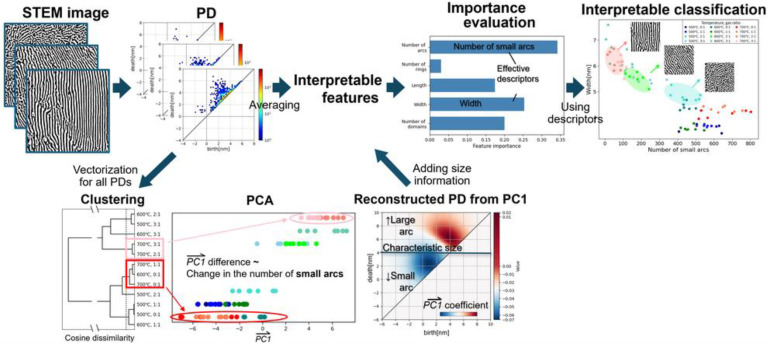
Conceptual diagram for extraction of interpretable structural features from STEM images by using PH and classification of nanostructures by using the determined effective features.

**Figure 2 nanomaterials-14-01413-f002:**
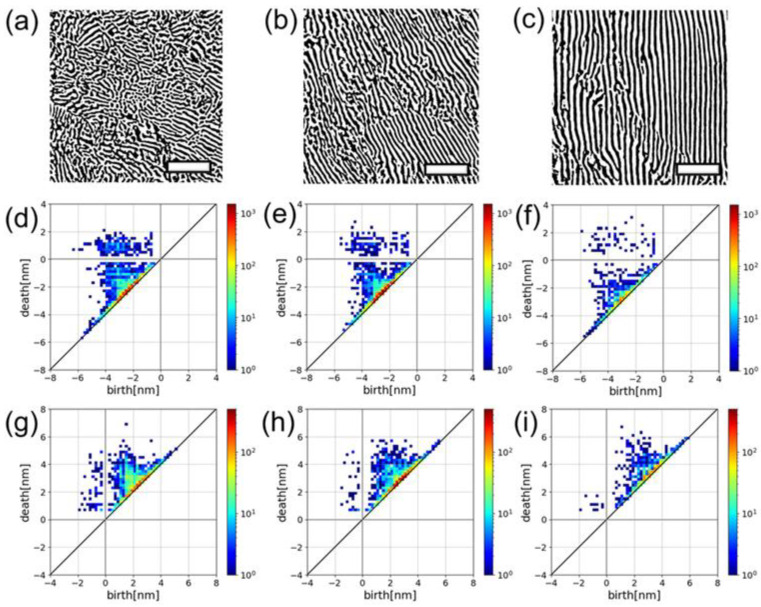
Binarized high-angle annular dark-field scanning transmission electron microscopy (HAADF-STEM) images of Pt-CeO_2_ nanostructures synthesized at (**a**) 500 °C, (**b**) 600 °C, and (**c**) 700 °C with a CO:O_2_ gas ratio of 2:1. Scale bars represent 100 nm. (**d**), (**e**), and (**f**) Zeroth and (**g**), (**h**), and (**i**) first persistent diagrams (PDs) of CeO_2_ phases in the nanostructures corresponding to (**a**), (**b**), and (**c**), respectively. The color bars indicate the number of *b*-*d* pairs.

**Figure 3 nanomaterials-14-01413-f003:**
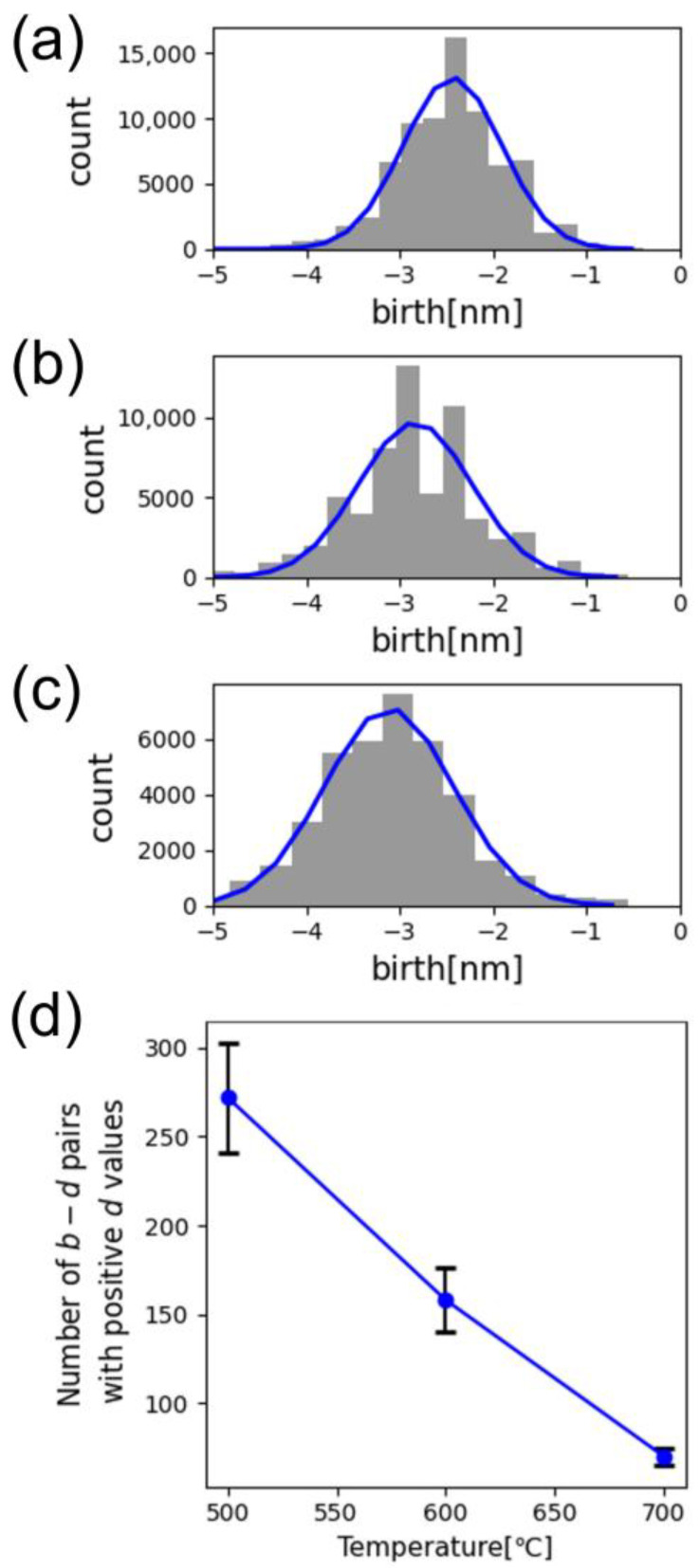
Sum of the frequency distributions of the number of *b*-*d* pairs from the seven images as a function of *b* determined from zeroth PDs of nanostructures synthesized at (**a**) 500 °C, (**b**) 600 °C, and (**c**) 700 °C with a CO:O_2_ gas ratio of 2:1. The blue solid lines indicate the Gaussian fitting at the center of each binary region. (**d**) Dependence of the number of *b*-*d* pairs with positive *d* values derived from the zeroth PDs of the nanostructures on synthesis temperature.

**Figure 4 nanomaterials-14-01413-f004:**
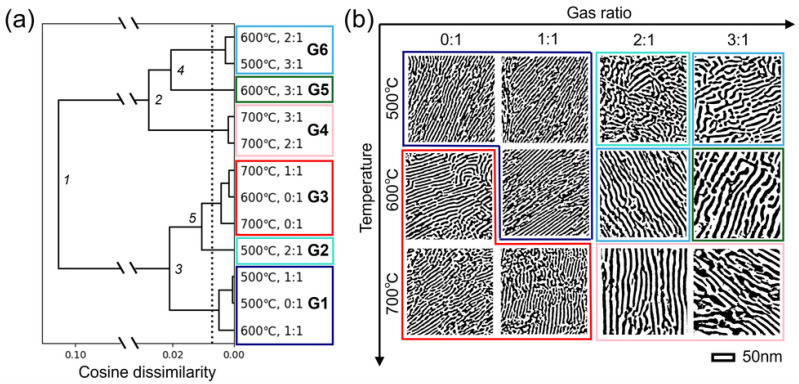
(**a**) Hierarchical clustering of Pt-CeO_2_ nanostructures with cosine dissimilarity calculated from zeroth vectorized PDs. The branch points are numbered as 1, 2, … in descending order of cosine dissimilarity. The dashed line is the threshold for clustering and is the maximum deviation calculated from the mean of cosine dissimilarity for the same set of synthesis conditions. The 12 nanostructures were finally clustered into six groups (G1–G6). (**b**) Binarized STEM images of Pt-CeO_2_ nanostructures surrounded by the colors corresponding to those of the six groups (G1–G6) in (**a**).

**Figure 5 nanomaterials-14-01413-f005:**
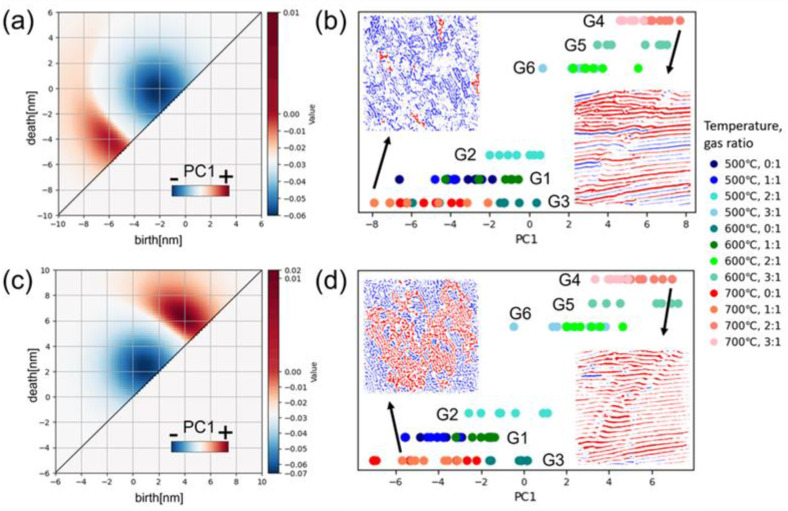
Zeroth (**a**) and first (**c**) reconstructed PDs of Pt-CeO_2_ nanostructures obtained from PC1 of PCA. The color bars indicate the PC1 coefficients. PC1 values from all 84 vectorized (**b**) zeroth and (**d**) first PDs categorized into six groups through hierarchical clustering. Inset images show the inverse analysis results obtained from the (**a**) zeroth and (**c**) first reconstructed PDs for the nanostructures synthesized at 700 °C with a CO:O_2_ gas ratio of 1:1 (left) and at 700 °C with a CO:O_2_ gas ratio of 2:1 (right). The blue and red regions in the image in (**b**) correspond to the nanostructures from the *b*-*d* pairs with −4 nm ≤
*b*, *d <* 0 and *b*
< −4 nm, *d* < 0 in the zeroth PDs, respectively, and those in (**d**) correspond to the structures from the *b*-*d* pairs with 0 <
*b*, *d*
≤ 4 nm and 0 < *b*, 4 nm ≤ *d* in the first PDs, respectively.

**Figure 6 nanomaterials-14-01413-f006:**
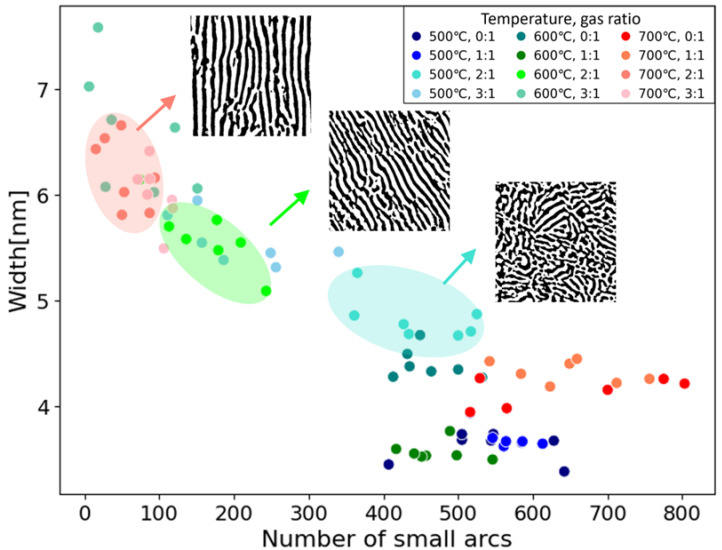
Scatter plot of the number of small arcs with the characteristic size and the width of the CeO_2_ phase for 12 Pt-CeO_2_ nanostructures fabricated at different annealing temperatures and gas ratios. The colors show that there are three well-classified groups that were fabricated at different temperatures but with the same gas ratio. Inset images are corresponding binarized STEM images for each group.

**Table 1 nanomaterials-14-01413-t001:** Summary of five interpretable features, corresponding to zeroth and first PD quadrants, and the estimated average and error (in parentheses) for the Pt-CeO_2_ nanostructures synthesized at different temperatures with a CO:O_2_ gas ratio of 2:1. Each average and error was calculated using standard deviation from the seven images of structures formed under the same synthesis condition.

Geometric Information	Representation in PD	500 °C CeO_2_	500 °C Pt	600 °C CeO_2_	600 °C Pt	700 °C CeO_2_	700 °C Pt
Number of domains	Number of *b*-*d* pairs in positive death region of zeroth PD	272 (31)	71 (11)	158 (18)	67 (7)	70 (5)	68 (9)
Width of stripes [nm]	Average of twice the value of absolute birth value in zeroth PD	4.83 (8)	5.04 (7)	5.61 (12)	6.21 (16)	6.21 (13)	6.91 (17)
Total length of stripes [nm]	Number of *b*-*d* pairs with short lifetime of zeroth PD	2805 (185)	3073 (172)	2458 (245)	2642 (231)	1466 (211)	1473 (217)
Number of rings	Number of *b*-*d* pairs in negative birth region of first PD	34 (8)	186 (29)	34 (3)	92 (14)	25 (7)	25 (3)
Number of arcs	Number of *b*-*d* pairs in positive birth region of first PD	550 (27)	446 (38)	390 (27)	352 (18)	203 (16)	193 (16)

## Data Availability

Data are contained within the article and Appendix A.

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
