# Peer review of "Interpretable Structural Evaluation of Metal-Oxide Nanostructures in Scanning Transmission Electron Microscopy (STEM) Images via Persistent Homology"

_nanomaterials, 2024, doi:10.3390/nano14171413_

Round 1

Reviewer 1 Report

Comments and Suggestions for Authors

This paper employs persistent homology to extract geometric features from high-angle annular dark-field STEM images of Pt-CeO2 nanostructures synthesized under different annealing conditions and varying gas ratios of CO and O2. The significance of this work lies in its potential to advance the field of materials science by providing a method to classify and understand nanostructures based on their geometric features. In this context, the authors succeed in their goal of extracting interpretable features to enable structural classification.

Although the importance of this field is clear, the subject is not easily interpreted by readers who are not experienced in this area. To facilitate reading, I recommend that the authors highlight in the introduction the significance of their results within the field of materials science. In particular, the authors should emphasize the importance of obtaining such a large and detailed amount of geometrical information from the nanostructures. They should clearly explain which functional properties of the examined nanomaterials, and nanomaterials in general, could be better elucidated by this approach.

Even if already published in their previous papers, I suggest, for completeness and clarity, including the original HAADF-STEM images in the supplementary information. This will provide readers with a direct reference to the raw data from which the geometric features were extracted.

Reviewer 2 Report

Comments and Suggestions for Authors

Dear authors and editors,

the presented manuscript describes the homogical evaluation HAADF-STEM images of Pt-CeO2 nanostructures formed under different preparation conditions (temperature and gas ratio) in order to extract differences and key features determining the physical properties of the nanostructures. I like the idea of analysing aquired images beyond the simple measurement of feature sizes. Therefore I think the manuscript should be published after correction of some issues:

- Although I think it is described in one of the authors' earlier publications: Can you comment on how the binarization of the images is
 done (How the threshold was chosen?)

- Figure 1 d)-i): What is the quantity displayed as color in the persistent diagrams?

- Page 4: It is stated that the average width of the CeO2 is increasing with temperature as the number of values of the b is increasing. However, the position of of the gaussian in the histograms in Fig2. a) and b) is not changing, only to c). What is the error of b and thus of the stripe width?

- Table 1: How are the errors determined? (Are they the standard deviation over the 7 images from each sample?)

- I have a problem in understanding the result table 1: On the one hand the number of domains of CeO2 is decreasing with increasing temperature. At the same time the length of the stripes is decreasing, but the width of the stripes is approximately constant (compared to changes in length). I would have expected that the length of the stripes need to increase when the number of domains goes down. Also decr
easing stripe length (table 1) contradicts the images in figure 1a-c.

- Figure 4a) and c) What is "value"? It should be the same as in Fig. 1 d)-i), but the "values are completly different (a small change I can understand as it is the reconstruction only from the first principle component (possibly it is because the component 0 is missing?).).

- Caption of Figure 4: I think the references to the subfigures are erroneous. It is written twice "(a) and (b)", but should be "(a) and (c)"

Reviewer 3 Report

Comments and Suggestions for Authors

The authors evaluated Pt-CeO2 samples series prepared under various conditions. Pt-CeO2 and similar materials can be used as effective catalyst, so prediction of their properties may be crucial for their application in CO2 reduction processes. Understanding the relationship between preparation conditions and structural characteristics may help to optimize these materials for specific applications.

Unfortunately, the authors analyzed just single area on each sample. To my opinion analysis of multiply areas within STEM sample, would make the interpretation more reliable.    

Beside this, I do not have any significant remarks or comments. 

Overall, the manuscript is written in logical and clear form, however, the scientific nature of the results and the impact on future sample preparation could be emphasized more.

Round 2

Reviewer 2 Report

Comments and Suggestions for Authors

Dear authors and editor,

thank you very much for considering most of my comments on the manuscript. I think, that the manuscript has improved by the
changes. However, I found that my last point was not addressed in the response letter, but became improved with the addition
 and explanation of new figures S1 and  S2. Also the added paragraph a the end of section 3.3 was good (possibly it was just
 forgotten to be added in the response letter).

So I think that the manuscript can be accepted for publication.